# Air Quality over China

**Gerrit de Leeuw** [1,2,3,4,*] , **Ronald van der A** [1,3] , **Jianhui Bai** [5] , **Yong Xue** [4] , **Costas Varotsos** [4,6] , **Zhengqiang Li** [2] , **Cheng Fan** [2] , **Xingfeng Chen** [2] , **Ioannis Christodoulakis** [6] , **Jieying Ding** [1] , **Xuewei Hou** [3] , **Georgios Kouremadas** [6] , **Ding Li** [4] , **Jing Wang** [3] , **Marina Zara** [1,†] , **Kainan Zhang** [7] and **Ying Zhang** [2]

1   KNMI (Royal Netherlands Meteorological Institute), 3730AE De Bilt, The Netherlands;
    ronald.van.der.a@knmi.nl (R.v.d.A.); jieying.ding@knmi.nl (J.D.); marina.zara@knmi.nl (M.Z.)
2   Aerospace Information Research Institute, Chinese Academy of Sciences (AirCAS), No. 20 Datun Road,
    Chaoyang District, Beijing 100101, China; lizq@radi.ac.cn (Z.L.); fancheng@radi.ac.cn (C.F.);
    chenxf@aircas.ac.cn (X.C.); zhang_ying@aircas.ac.cn (Y.Z.)
3   School of Atmospheric Physics, Nanjing University of Information Science & Technology (NUIST), No. 219
    Ningliu Road, Nanjing 210044, China; houxw@nuist.edu.cn (X.H.); wangjing2004y@nuist.edu.cn (J.W.)
4   School of Environment Science and Spatial Informatics, China University of Mining and Technology (CUMT),
    Xuzhou 221116, China; yxue@cumt.edu.cn (Y.X.); covar@phys.uoa.gr (C.V.); lidinf@cumt.edu.cn (D.L.)
5   LAGEO, Institute of Atmospheric Physics, Chinese Academy of Sciences (IAP, CAS), Beijing 100029, China;
    bjh@mail.iap.ac.cn
6   Department of Environmental Physics and Meteorology, University Campus, Bldg PHYS-V, National and
    Kapodistrian University of Athens, GR-157 84 Athens, Greece; ichristo@phys.uoa.gr (I.C.);
    gkoure@phys.uoa.gr (G.K.)
7   College of Geological Engineering and Geomatics, Chang'an University, Xi'an 710054, China;
    zhangkn@chd.edu.cn
*   Correspondence: gerrit.de.leeuw@knmi.nl
†   Current affiliation: Institut für Photogrammetrie und Fernerkundung (IPF), Karlsruher Institut für
    Technologie (KIT), 76128 Karlsruhe, Germany.

**Abstract:** The strong economic growth in China in recent decades, together with meteorological factors, has resulted in serious air pollution problems, in particular over large industrialized areas with high population density. To reduce the concentrations of pollutants, air pollution control policies have been successfully implemented, resulting in the gradual decrease of air pollution in China during the last decade, as evidenced from both satellite and ground-based measurements. The aims of the Dragon 4 project "Air quality over China" were the determination of trends in the concentrations of aerosols and trace gases, quantification of emissions using a top-down approach and gain a better understanding of the sources, transport and underlying processes contributing to air pollution. This was achieved through (a) satellite observations of trace gases and aerosols to study the temporal and spatial variability of air pollutants; (b) derivation of trace gas emissions from satellite observations to study sources of air pollution and improve air quality modeling; and (c) study effects of haze on air quality. In these studies, the satellite observations are complemented with ground-based observations and modeling.

**Keywords:** remote sensing; China; atmospheric composition; air quality; aerosols; trace gases; emissions; time series; material degradation

## 1. Introduction

The air quality in China has deteriorated for many years, resulting in serious air pollution problems. This is due to different factors, both natural and anthropogenic. The strong economic growth with the establishment of large industrialized areas, which in turn stimulated urbanization leading to a high population density, has contributed to the air pollution problems experienced in the late 20th and early 21st centuries. However, not only anthropogenic factors need to be considered in studies on the air quality in China, which is also influenced by meteorological factors (cf. [1,2] and references cited therein), as well as

natural emissions, such as wind-generated dust [3,4], emissions from wild fires [5], and biogenic emissions [6]. Furthermore, the location of urban centers, such as the Sichuan Basin with Chengdu and Chongqing [7] and the Guanzhong Basin (Xi'an) [8], both situated between mountains preventing the transport of pollutants out of the area, lead to the accumulation of pollutants. This accumulation is further promoted by local meteorological factors conducive to the formation of haze and the transport of pollutants, which are known to occur in particular over the North China Plain (NCP) [9,10]. In addition, long range transport influences air quality [11,12]. A well-known example is the occurrence of high concentrations of dust over the Beijing area, due to transport of dust generated during storms over deserts in the north and west of China [3,4]. Large-scale weather systems, such as the East Asia summer monsoon [13] and the Siberian High [12,14] influence the air quality in different parts of China in different ways.

Poor air quality leads, among other things, to health problems and reduced life expectancy [15], reduction of solar radiation reaching the surface [16], reduction of visibility [17], deterioration of cultural heritage [18,19], and it influences agricultural yields and water quality (e.g., [20,21]). To reduce the concentrations of pollutants, air pollution control policies have been successfully implemented by the Chinese government, resulting in a gradual decrease of the concentrations of atmospheric pollutants during the last decade, as evidenced by both satellite and ground-based measurements (see [22,23] and references cited therein).

Long-term data records, which can be used for air quality studies in China, are sparse. A national network of ground-based air quality monitoring stations was established in 2013 and provides concentrations of $PM_{2.5}$, $PM_{10}$, $SO_2$, $NO_2$, $O_3$, and CO. This network is maintained by the China National Environmental Monitoring Center (CNEMC) of the Ministry of Ecology and Environment (MEE; for more details, see http://www.mee.gov.cn/, last access: 26 April 2021). Hourly and 24-h moving averages for each site or city are used in many air quality studies [24–26].

Ground-based remote sensing data are available from broadband solar radiation measurements. Broadband solar radiation has been measured in China by the China Meteorological Administration (CMA) since the 1950s at 98 sites [27], which evolved into the Chinese national solar radiation network with 14 stations [28], providing continuous data since 1993. Such measurements do not provide direct information on air quality but can be used to derive long time series showing the effect of atmospheric pollution on the amount of solar radiation reaching the surface [29,30].

Sun photometers measure direct solar radiation in a number of wavebands from the UV to the NIR (depending on the type of instrument), as well as scattered radiation, which together can be used to retrieve aerosol macroscopic and microscopic properties integrated over the atmospheric column (for detailed descriptions of instruments and protocols, see https://aeronet.gsfc.nasa.gov/, last access 26 April 2021). The first sun photometers in China were installed in Beijing and Xianghe, as part of the National Aeronautics and Space Administration (NASA) Aerosol Robotic Network (AERONET) [31], in 2001. Complementary to AERONET, and in part shared with AERONET, Chinese networks have been developed [32], such as CARSNET (China Aerosol Remote Sensing Network), established by the China Meteorological Administration (CMA) [33,34] (50 sites), and SONET (Sun-Sky Radiometer Observation Network) established by the Institute of Remote Sensing and Digital Earth, Chinese Academy of Sciences (RADI/CAS), Beijing [35] (16 sites). CARE China (Campaign on Atmospheric Aerosol Research network of China) [36] includes 36 sites where handheld sun photometers are used (providing only direct sun measurements) together with ground-based instrumentation providing information on chemical composition.

Whereas sun photometers are passive instruments providing column-integrated aerosol properties, LiDARs (light detection and ranging) are active instruments which provide aerosol extinction and backscatter at one or two wavelengths, as a function of

height in the atmospheric boundary layer. In China, LiDARs are operated by individual groups but to our knowledge there are no organized operational networks.

MAX-DOAS (Multi-Axis Differential Optical Absorption Spectroscopy) observations of $NO_2$ are made at several institutes in China, most notably the Anhui Institute of Optics and Fine Mechanics, which operates a MAX_DOAS network in the Yangtze River Delta. Other routine MAX-DOAS observations are done at IAP (Institute of Applied Physics), Beijing and at NUIST (Nanjing University of Information Science and technology), Nanjing.

Satellite remote sensing provides information on aerosols and trace gases over large spatial scales across the whole planet, with global coverage from sensors with wide enough swath width, such as OMI (Ozone Monitoring Instrument) and TROPOMI (TROPOspheric Monitoring Instrument) for trace gases and MODIS (Moderate Resolution Imaging Spectroradiometer) and VIIRS (Visible Infrared Imaging Radiometer Suite) for aerosols. The information obtained from each sensor is consistent in the sense that the same instrument and processing technique are used across the whole planet, whereas the information may be different when different sensors are used or even when different algorithms are applied to data from the same sensor. For the evaluation of information obtained from satellite-based sensors, independent ground-based reference data, such as those obtained from the sun photometer networks mentioned above, are indispensable. The information which can be obtained on atmospheric trace gases and aerosols depends on the sensor characteristics and has been evolving since the first use of spaceborne instruments for this purpose, in 1978 [37]. A spectrometer like GOME (Global Ozone Monitoring Experiment, on board ERS-2; 1995–2003) was originally designed for the retrieval of $O_3$, at wavelengths in the UV-VIS part of the electromagnetic spectrum. However, the spectra also contain information on other atmospheric constituents, like $NO_2$ and $SO_2$, and later instruments were progressively designed for the retrieval of more trace gases and greenhouse gases from the UV-VIS (ultraviolet-visible) and NIR (near-infrared) parts of the spectra. These later spectrometers include SCIAMACHY (SCanning Imaging Absorption spectroMeter for Atmospheric CartograpHY, ENVISAT; 2002-2012), OMI (AURA; 2004-present), GOME-2 (METOP; 2006-present) and TROPOMI (Sentinel 5a; 2017-present). In addition, technological advances resulted in the decrease of sub-satellite pixels from 300 km for GOME to 7 km for TROPOMI.

Spectrometers are sometimes also used to obtain aerosol information such as the absorbing aerosol index (AAI) or aerosol layer height, but radiometers are more suitable for aerosol retrieval because better spatial resolution can be achieved which benefits the separation of cloud-contaminated from clear pixels, a basic requirement for reliable aerosol retrieval. In contrast to discrete absorption bands used for trace gas retrieval, the aerosol spectrum varies smoothly with wavelength and therefore radiometers can be used with a limited number of discrete wavebands, preferably with wavelengths across the whole spectrum between the UV and the TIR (thermal infrared). In addition, two or more viewing directions and detection of polarization are beneficial for aerosol retrieval. Sensors designed for aerosol retrieval are MODIS (on TERRA; 1999-present; on AQUA; 2002-present) and its follow-up VIIRS (Suomi-NPP; 2011-present), which are single view instruments with wavelengths from the UV to the TIR, MISR (Multi-angle Imaging SpectroRadiometer, TERRA, 1999-present) with wavelengths in the VIS/NIR but with 7 viewing angles, and the POLDER instruments (POLarization and Directionality of the Earth's Reflectances, on three different satellites, the last one being POLDER-3 on PARASOL, 2005–2013) series with many more viewing angles and polarization information. However, radiometers designed for purposes other than aerosol retrieval may also be suitable for this purpose. In this paper, the Along Track Scanning Radiometer (ATSR-2, ERS-2, 1995–2003) and the Advanced ATSR (AATSR, SCIAMACHY, 2002–2012), together referred to in this paper as ATSR, are used for aerosol retrieval and analyzed together with MODIS data. In addition, data from the Cloud-Aerosol LiDAR with Orthogonal Polarization (CALIOP, CALIPSO; 2006-present) are used to study the 3-D variation of aerosol properties. All satellites mentioned above are in a sun synchronous orbit, i.e., they view each location on Earth once per day (or less,

depending on swath width). In contrast, geostationary satellites view only a specific part of the planet, but with high frequency. In this paper some results are presented from the retrieval of aerosol information using data from the Advanced Himawari Imager (AHI, HIMAWARI-8; 2014-present).

The Dragon program is a cooperation between ESA and the Ministry of Science and Technology (MOST) of the People's Republic of China which started in 2004. There have been four phases, each lasting four years (http://dragon4.esa.int/, last visited 26 April 2021). Dragon 4 focuses on the scientific exploitation of ESA, ESA Third Party Missions, Copernicus Sentinels and Chinese Earth observation (EO) data for geo-science and applications development. In this paper we present an overview of the Dragon 4 (2016–2020) project "Air Quality over China" (AQ-China) which is part of the topic "Atmosphere, climate & carbon cycle". Contributions to this project include the use of information on trace gases and aerosols obtained from satellite observations, which contribute to our knowledge on atmospheric composition and effects on air quality. OMI, GOME-2, and TROPOMI data were used to obtain information on the spatial distributions of concentrations and emissions of trace gases, $SO_2$ and $NO_2$, over China. Emissions were estimated using satellite data in the top-down approach [22], as opposed to the traditional bottom-up statistical approach which provides data that become outdated quickly and do not reflect the real air pollution status in many regions. The work by van der A et al. [22] was extended to 2018 and studies were made on the detection of the emission of $NO_2$ by ships along the Chinese coast [38] and on the Yangtze River. Emission sectors were identified by combining emissions of $SO_2$ and nitrogen oxides ($NO_x$ = NO + $NO_2$) derived from OMI measurements of $SO_2$ and $NO_2$ (Zara et al., in preparation). In this paper the status of emissions of $NO_x$ and $SO_2$ over the last twelve years, derived from satellite observations, will be presented.

The work by de Leeuw et al. [4] and Sogacheva et al. [23,39] on the spatiotemporal variation of the aerosol optical depth (AOD) over China for the period 1995–2017, including the use of ATSR and MODIS data to construct an AOD time series, is extended to 2020. As part of the Dragon 4 cooperation, the ATSR time series was extended back in time to 1983 using AVHRR data [40]. MODIS data was also used together with CALIOP data to study the 3-D variation of aerosol properties over the Beijing–Tianjin–Hebei area (BTH), the Yangtze River Delta (YRD) and the Pearl River Delta (PRD) [41].

The time series of $SO_2$, $NO_2$ and AOD show the initial increase of the concentrations early in the 21st century, as well as the progressive effects of the measures by the Chinese government to improve the air quality over China. In addition, the AOD data can be used to obtain information on aerosol concentrations near the surface, which in air quality studies are expressed as $PM_{2.5}$: the integrated mass of dry aerosol particles, sampled under ambient conditions with an aerodynamic diameter of 2.5 µm (i.e., wet aerosol). Studies to provide $PM_{2.5}$ from AOD data have been undertaken using both sun photometer and satellite data, together with physical models [2,42] or a neural network [43]. In addition, studies were made on the effects of long-range transport of aerosol on $PM_{2.5}$ versus the effect of local production, and on the effects of large-scale meteorology on transport pathways [12,44].

In response to the lockdown measures to contain the COVID-19 virus outbreak, in early 2020, the concentrations and emissions of trace gases and aerosols in China changed substantially, which provided a sad but also unique opportunity to study the effect of a drastic change in anthropogenic emissions on air quality. Satellite observations were widely used and reports appeared in the peer-reviewed scientific literature. Although this was not among the main goals, several partners of the AQ-China project were involved and the results published in [26,45–47] show estimates of the reductions of both concentrations and emissions of pollutants over China and worldwide.

As an application of the satellite data studied in the AQ-China project, they were used in studies on the effects of air pollution and other environmental parameters on materials at the National and Kapodistrian University of Athens [19,48]. The results from this study are briefly reported in this paper as a part of the AEROSOL sub-project.

The Dragon series of projects is mainly focused on satellite remote sensing. However, satellite measurements cannot be seen separate from ground-based observations, which are needed for calibration of the satellite sensors, as well as for process studies. Furthermore, satellite and ground-based observations are complementary and provide different types of information. As a contribution to the Dragon project, 10 years of data from ground-based measurements of solar radiation (direct and diffuse) in different wave bands and Photosynthetically Active Radiation (PAR), fluxes of biogenic volatile organic compounds (BVOCs), $O_3$ concentrations and meteorological parameters, at four forest sites in different climate zones in China, were used together with satellite measurements of aerosols and trace gases to formulate an empirical model of BVOC emissions (EMBE). The EMBE model is evaluated at different time scales (weekly to annual). This work is important because VOCs, which can be of anthropogenic and biogenic origin, play an important role in the oxidizing capacity of the atmosphere and thus in chemical reactions involving $O_3$ and NOx [49,50]. VOCs are also important precursors for the secondary formation of aerosol particles [51]. The results of this study were used to propose suggestions for a way to reduce the concentrations of air pollutants such as $PM_{2.5}$ and $O_3$ [52]. Understanding of the formation of new aerosol particles (new particle formation or NPF) over forest regions [51,53] through gas to particle conversion has contributed to the development of satellite based methods to derive NPF with global application [54,55].

It is noted that this paper was prepared after the end of the DRAGON 4 project, when most results had already been published. In this paper an overview is presented of work performed between the DRAGON 4 AQ-China partners, including a summary of results and references where all details can be found. Few new results, extending earlier findings to the end of the project, are included as well.

## 2. The Dragon 4 Sub-Project "Air Quality over China" (AQ-China)

The Dragon 4 project is the fourth of a series of four-year projects which started in 2004 (for detail on Dragon 4 and earlier projects, see http://dragon4.esa.int/, last visited 26 April 2021). The Dragon 4 phase started in 2016 and ended in the summer of 2020. Dragon 4 included 8 topics and the sub-project Air Quality over China (AQ-China) was part of the topic "Atmosphere, climate & carbon cycle". The main theme of this project was the study of air pollution over China using satellite observations, to achieve a better understanding of the sources of air pollution and their spatial and temporal variabilities.

### 2.1. List of Sub-Projects and Teaming

Air quality is determined by a number of atmospheric constituents, including trace gases ($NO_2$, $SO_2$, $O_3$ and CO) and aerosols, and in particular their concentrations near the surface. These surface concentrations are not directly available from satellites. Yet, satellite data provide important information on the spatiotemporal variations of atmospheric constituents. As discussed in the Introduction section, trace gases and aerosols are measured from space using sensors with very different characteristics and retrieval methods. Therefore, usually different research groups focus on different species, and combine their expertise in programs like the Dragon series. This strategy was also followed in the AQ-China project which consisted of two sub-projects:

- Air Quality Observations and Emission Estimates.
- AEROSOL: Satellite-derived aerosol properties over Mainland China: application to air quality and trend analysis.

A third sub-project, on the assessment of the characteristics, sources and impact of haze in China was not carried out, but part of this work was included in the AEROSOL sub-project and results were published in Li et al. [56].

### 2.2. Description and Summary Table of EO and Other Data Utilized

The focus of the Dragon 4 project AQ-China was on the use of satellite data for air quality (AQ) studies in China, but, as explained in the Introduction, ground-based in situ

and remote sensing data have also been used as a necessary part for validation and as a source for complementary information. The data used are summarized in Table 1, the instruments were briefly described in the Introduction.

**Table 1.** EO and other data utilized in the AQ-China project.

| Satellite Data | | | | |
|---|---|---|---|---|
| Type of Data | Species | Instrument | Satellite | Period Available |
| Aerosol | AOD | ATSR-2 | ERS-2 | 1995–2003 |
| | | AATSR | ENVISAT | 2002–2012 |
| | | AVHRR | Several platforms | 1983–present |
| | | MODIS | Terra | 2000–present |
| | | | Aqua | 2002–present |
| | | VIIRS | S-NPP | 2011–present |
| | AOD Aerosol type Vertical profile | CALIOP | CALIPSO | 2006–present |
| Trace gas | $NO_2$, $SO_2$ | OMI | Aura | 2004–present |
| | | TROPOMI | Sentinel-5 | 2017–present |
| Temperature | | MODIS | Terra/Aqua | 1999/2002–present |
| Humidity | | AIRS | Aqua | 2002–present |
| Ground-Based Data | | | | |
| Type of data | Species | Data source | | Period available |
| Aerosol | $PM_{2.5}$ | MEE AQ monitoring network | | 2013–present |
| | | Rp1400a | Xinglong | 2008–present |
| | AOD, AE *, and other retrieved aerosol properties | Sun Photometer networks: AERONET (public) CARSNET SONET CARE-China | | Dates vary by site |
| Trace gases | NO, $NO_2$, $SO_2$, $O_3$ | Gas analyzer | Xinglong | 2005–present |
| | BVOCs | REA, GC-FID/MS | Typical forests | Dates vary by site |
| Meteorological Data | | | | |
| Solar radiation | Global, direct, diffuse, UV, etc. | Solar radiation sensors | Xinglong | 2005–present |
| Temperature, humidity, wind speed | | Weather station | Xinglong | 2006–present |

* AE is the AOD Ångström Exponent.

## 3. AQ-China Aims and Approach

AQ-China consisted of two sub-projects, as presented in Section 2.1, each with their own aims and approach.

### 3.1. Air Quality Observations and Emission Estimates

The subproject Air Quality Observations and Emission Estimates in practice consisted of two parts. One part predominantly used satellite observations and a model to determine $NO_2$ and $SO_2$ emissions. The emphasis of the other part was on field campaigns in different climatic zones in China providing ground-based observations which, together with satellite-retrieved parameters, are used to formulate empirical emission models.

### 3.1.1. Research Aims

Top down Emission Estimates of $NO_x$ and $SO_2$

As the number of Chinese megacities is steadily increasing, air pollution issues are brought at the forefront of public awareness. In the aftermath of the progress of the Chinese economy and social welfare, the emissions of air pollutants estimated using the traditional bottom-up statistical approach become quickly outdated and do not reflect the real air pollution status in many regions, whereas large discrepancies between national emission estimates for China as a whole and the sum of the regional estimates of its provinces underline the need for alternative ways of emission estimation. In this sub-project, we focused on the status of emissions of $NO_x$ and $SO_2$ in the last twelve years, as derived from satellite observations. In particular, the aims of this research were the use of satellite data (OMI, TROPOMI) for top-down estimates of the emissions of $NO_x$ and $SO_2$ over China, the analysis of the spatial distributions of the concentrations and emissions of these trace gases and their temporal variation to estimate the effects of measures to improve air quality.

Emission of Volatile Organic Compounds (VOCs)

Volatile organic compounds (VOCs) play an important role on the oxidative capacity of the atmosphere and in the photochemical reactions involving pollutants, such as $NO_2$, $O_3$ and secondary organic aerosol (SOA). Terrestrial vegetation is the dominant source of biological volatile organic compounds (BVOCs) in the atmosphere [57]. The driving factors for BVOC emissions are solar irradiation (PAR) and air temperature. Solar irradiation near the surface is affected by the scattering and absorption of light by gases and aerosols in the atmospheric column. The research aims of this part of this subproject were (a) to investigate effects of photochemical transformation on the formation of pollutants in Northern China; (b) to formulate an empirical model for BVOC emissions (EMBE) in terms of solar irradiation, meteorological parameters and concentrations of atmospheric constituents, and (c) to develop an empirical model of global solar irradiance at the ground and at the top of the atmosphere (TOA).

### 3.1.2. Research Approach

Space-Based Trace Gas Emissions in China

*NOx emissions*

The DECSO algorithm [58] is specifically designed to use daily satellite observations of column concentrations for fast updates of emission estimates of short-lived atmospheric constituents on a mesoscopic scale ($0.25° \times 0.25°$). We used the DECSO algorithm together with the regional chemistry transport model CHIMERE [59] with a resolution of $0.25°$, driven by operational meteorological forecast of the European Centre for Medium-Range Weather Forecasts (ECMWF). We used $NO_2$ observations from the OMI instrument [60]. Tropospheric $NO_2$ column retrievals were calculated with the QA4ECV algorithm [61]. This data is available through the Tropospheric Emission Monitoring Internet Service (TEMIS) portal (http://www.temis.nl/, last access 26 April 2021). $NO_2$ retrievals at cloud fractions larger than 20% were filtered out to reduce the influence of the modelled $NO_2$ column below the clouds. Retrievals with low clouds (below 800 hPa) were also rejected because the intersection of the cloud with the $NO_x$ bulk makes the retrieval too sensitive to the exact cloud height. Before comparing the model simulations with the satellite observations, the CHIMERE vertical profiles were extended from the model ceiling (at 500 hPa) to the tropopause with a climatological partial column. The profiles were then interpolated to the observational footprints (having a lower spatial resolution), after which the averaging kernel can be directly applied. More details can be found in Mijling and Van der A [58] and Ding et al. [62]. The algorithm has been validated for the region of South East Asia in Ding et al. [63].

*SO2 emissions*

We used the $SO_2$ observations of the OMI satellite instrument as derived by Theys et al. [64]. The retrieval method is based on a Differential Optical Absorption Spectroscopy (DOAS) scheme to determine the slant columns from measured spectra in the 312–326 nm spectral range.

A trend study has been performed based on the monthly averaged OMI $SO_2$ observations. The year-to-year variation has been derived for each province in China for the period 2005–2018. Using the derived provincial variability for scaling the a priori emission inventory MEIC of 2010 (our reference year), reliable monthly emissions are provided for the whole period 2005–2018 [22].

*Emission of volatile organic compounds (VOCs)*

For EMBE model development, BVOC emission fluxes were measured using a relaxed eddy accumulation (REA) and a gradient method. Solar radiation in three wavebands and PAR were measured using a solar radiation system. These data were used to derive the global horizontal irradiance and the direct and diffuse normal irradiances. $O_3$ concentrations near the surface were measured using an ozone monitor (Model 205, 2B Technologies Inc., Boulder, CO, USA). Meteorological variables (temperature, relative humidity) were measured using a HOBO weather station (Model H21, Onset Company, Bourne, MA, USA). These measurements were made in a subtropical *Pinus* forest in China from May 2013 to December 2016, as described in detail in Bai et al. [65].

Data collection from long term campaigns started in 2005 [55]. Ground-based radiation and meteorological data at four sites in Northern China were combined with OMI-derived vertical column densities of $NO_2$, $SO_2$, $O_3$ and HCHO and with AOD data derived from AATSR using FMI's ATSR dual view algorithm (ADV) [66,67]. The four sites are Yucheng and Luancheng, both representative for agricultural conditions, Xianghe, strongly influenced by urban conditions, and Xinglong which is an atmospheric background site. All four sites are operated by the Chinese Academy of Sciences (CAS). At Xinglong, also in situ measurements are made of concentrations of NO, $NO_2$, $SO_2$, $O_3$ and $PM_{2.5}$. For more detail, see Bai and Hao [68].

## 3.2. AEROSOL: Satellite-Derived Aerosol Properties over Mainland China: Application to Air Quality and Trend Analysis

The AEROSOL sub-project included two parts. The first part was the use of satellite data to study the spatiotemporal variation of aerosols over China. The second part was a study on the effects of pollutants on materials.

### 3.2.1. Research Aims

The initial focus of the AEROSOL subproject was on algorithm development for aerosol retrieval over China using ATSR data and the application of the results to the analysis and interpretation of air quality and trends. With the ATSR era from 1995–2012, this period was extended with MODIS data to the present (2020) and with AVHRR data to go back in the past until 1987. AOD maps were produced for many years showing both the variability of aerosol concentrations over China and the temporal variation, including the spatial distributions of the AOD trends for different periods of time. For the interpretation of the AOD data and their spatiotemporal variation, several studies were conducted in addition to the initial goals. These include the use of geostationary data, providing information on the diurnal variation of aerosols, and the application of machine learning techniques to obtain aerosol information from satellite data. Furthermore, for the application of satellite-derived aerosol data in AQ studies, the fine mode aerosol fraction is important. However, over land the fine mode fraction is currently not routinely available from satellite observations. Therefore, new work was undertaken to improve the use of satellite information for the purpose of estimating fine mode fraction and concentrations of particulate matter ($PM_{2.5}$). Another application is the effect of air pollution on materials, which was among the original aims of the AQ-China project.

### 3.2.2. Research Approach

Satellite Data

Satellite aerosol data were obtained from several platforms. The original aim was to use ATSR-2 and AATSR data, together providing 17 years (1995–2012) of AOD and Aerosol Exponent (AE) globally. The ATSR Dual View algorithm [66] for aerosol retrieval over land has been continuously improved as part of the ESA climate change initiative (cci) project Aerosol_cci [69,70] and the aerosol products were used for the production of aerosol Climate Data Records [71]. For the extension of this data set until the present, and to fill the gap between AATSR and its follow-up SLSTR (2016–present), data from MODIS/Terra were used because of the almost simultaneous overpasses of Terra and ENVISAT (10:30 and 10:00 a.m. local time, respectively). The ATSR and MODIS C5.1, C6.0 and C6.1 AOD products over China were evaluated by comparison with AERONET [4]. The biases of MODIS C6.1 and ATSR, with respect to AERONET AOD, of +0.07 and −0.07, respectively, shows that these products are suitable for merging during overlapping periods, with a similar adjustment of the MODIS data after 2012 and the ATSR-2 data before 1999 [4,23] and thus create multidecadal AOD time series from 1995 until 2017.

In addition, as part of the AQ-China project, the time series was extended backward to 1987 over relatively small areas in China and Europe, using data from AVHRR sensors. To this end, the RADI/CAS aerosol optical depth over land (ADL) algorithm [72] was further developed and evaluated versus AERONET and CARSNET data available over the study area. However, these reference data were not available before 2000 and therefore AOD derived from solar radiation measurements, using the broadband extinction method (BEM) as briefly described in the Introduction section, were used for validation [40].

In this paper, the MODIS/Terra and MODIS/Aqua AOD product MCD19A2, retrieved with the multiangle implementation of atmospheric correction (MAIAC) algorithm [73,74] was used to produce a MODIS AOD time series from 2011 to the end of 2020. The MAIAC MCD19A2 AOD product was used instead of the C6.1 DBDT product used by Sogacheva et al., [23,39] because of its better performance over China [75].

VIIRS AOD data were used in a study on the spatiotemporal variation of $PM_{2.5}$ in the Guanzhong basin. In this study, we used VIIRS EDR AOD at 550-nm wavelength with QF = 3 (https://www.bou.class.noaa.gov/saa/products/welcome; last access 26 April 2021), which were evaluated with SONET sun photometer data [42].

The integrated time series technique proposed by Mei et al. [76] was used to develop an improved aerosol retrieval algorithm (ITS) using data from the geostationary satellite Himawari-8 (H8), developed by the Japan Meteorological Agency (JMA). The advanced Himawari imager (AHI) onboard H8 is a 16-channel, multispectral imager which provides full-disk observations over East Asia and the western Pacific at 10 min intervals. For aerosol retrieval, the L1B TOA albedo values at 470, 510, 640, 870, and 2250 nm, measured at 10 min intervals from 09:00 a.m. to 04:00 p.m. China Standard Time (CST, i.e., UTC + 8:00; CST is local time used all over China), were used with a resolution of 5 km. These data are available from the JAXA website (http://www.eorc.jaxa.jp/ptree/index.html; last access: 26 April 2021). Real-time L2 cloud products with the same spatial–temporal resolution are used to select cloud-free pixels.

The retrieval of aerosol information from satellite data is an under-determined problem because of the limited number of degrees of freedom. Recently, studies have been published on the use of neural networks to obtain aerosol information from satellite data, by training them with information available from satellite sensors, complemented with information from other sources. Neural networks do not provide new information, or information which was not included in the training set, but the application is more efficient and faster, and often provides statistically more reliable results. In the AQ-China project a Convolutional Neural Network (CNN) was developed for the joint retrieval of AOD and FMF (fine mode fraction). The CNN was trained with MODIS and AERONET data and the results were evaluated with an independent reference data set (not used for training). For more detail, see Chen et al. [43].

Ground-Based Data

As mentioned above, ground-based reference data were used to validate and evaluate satellite-derived AOD products and the major sources for these data were the Chinese sun photometer networks SONET and CARSNET, as well as the AERONET sites in China. Brief descriptions and references to detailed information were presented in the Introduction section and will not be repeated here. This also applies to AOD derived from solar radiation measurements. Sun photometer data were also used in other studies on the use of AOD to derive $PM_{2.5}$ [2] and aerosol composition [77], as well transport pathways of aerosols.

Ground-based AQ data were used, i.e., $PM_{2.5}$ and concentrations of $NO_2$, $SO_2$, $O_3$ and CO, which are available from the national network of ground-based air quality monitoring stations in China established in 2013, see the Introduction section for more detail. In addition, local air quality monitoring networks were used.

Modeling

For the estimation of the spatiotemporal variations of $PM_{2.5}$ concentrations using VIIRS-derived AOD over the Guanzhong Basin [42], a two-stage spatiotemporal statistical model was used. Apart from VIIRS AOD data as the main variable, meteorological factors, land-cover, and population data were used as auxiliary variables. The two stages included a linear mixed effects (LME) model in stage 1, which was improved by using a geographically weighted regression (GWR) model in stage 2 (see Zhang et al. [42] for more detail).

Atmospheric Corrosion Athens Station (ACAS)

Since the end of 2002, a specialized corrosion station has been installed in Athens, Greece to investigate the corrosion and soiling of many different materials due to environmental factors such as temperature and humidity, as well as due to air pollution [18,19,78–81]. Figure 1 shows a part of the station's exposure facilities and Table 2 provides some information on the location of the Atmospheric Corrosion Athens Station (ACAS). Some of the experiments performed at ACAS are dedicated to the development of appropriate corrosion modeling techniques.

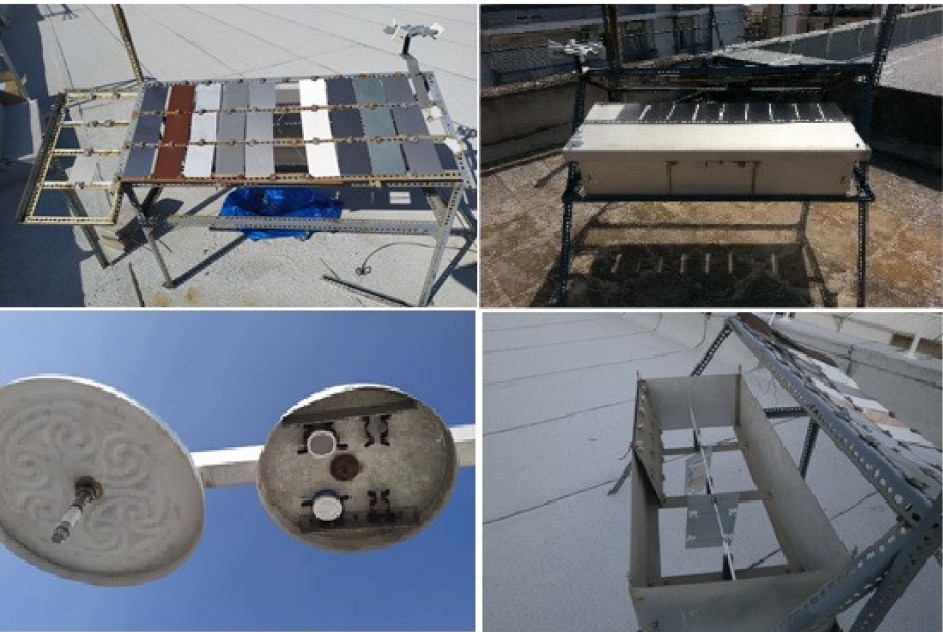

**Figure 1.** Views of the ACAS exposure site equipped with a metal rack on the roof of a building located in the Athens center, facing south. The rack consists of an inclined plane, for displaying samples of material under unsheltered conditions, and an aluminum box with an open bottom, for the exposure of materials' specimens under sheltered conditions. In addition, there is a mast with an arm that holds two rain shields, in the form of discs, under which passive particle collectors and diffusive samplers for gaseous pollutants are exposed [80,81].

**Table 2.** Critical information about ACAS.

| Site Name | Atmospheric Corrosion Athens Station (ACAS) |
|---|---|
| Country, region | Greece, Athens |
| Atmosphere | Urban traffic |
| Location (GPS) | 37°59′16″N, 23°43′39″E |
| Altitude | 90 m |
| Address | Aristotelous 17, 104 33, Athens, Greece |
| Description | Roof of a 7-floor building located near the centre of Athens, capital of Greece, with about 5 million inhabitants. |

A modern tool for estimating the effects of air pollution and other environmental parameters on materials is the Dose Response Function (DRF). The DRF expresses the relationship between the corrosion or deterioration of a material after one year of exposure and the levels or loads of pollutants in combination with climatic parameters. DRFs commonly use ground-based environmental data as input. In this work, the DRFs given in Kucera et al. [82] have been applied in the "traditional way" using ground-based environmental data collected at 10 European test sites during different experimental exposure periods and the results are compared with results from the satellite data processing procedure described in Christodoulakis et al. [48] for the same sites and periods.

## 4. Research Results and Conclusions

### 4.1. Air Quality Observations and Emission Estimates

#### 4.1.1. Results

Status of NOx and $SO_2$ emissions in China

Figure 2 illustrates an example of the spatial distribution of NOx emissions as derived from OMI measurements with the DECSO algorithm. Hot spots of NOx emissions are situated near large cities and along large rivers where industry is often located. Over sea a ship track is visible along the coast. Ship emissions and their trends are derived with DECSO and reported in Ding et al. [62].

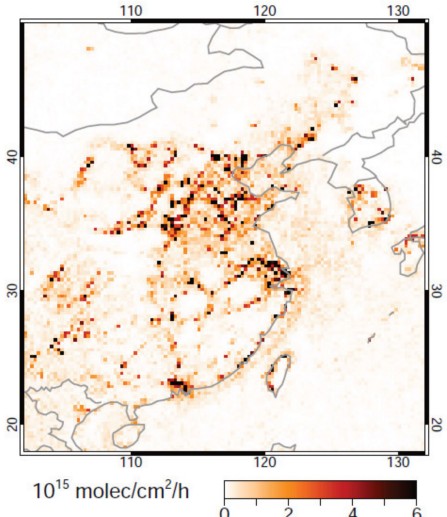

**Figure 2.** Annual distribution of space-based NOx emissions according to the DECSO v5 algorithm applied to OMI observations. This image shows the average situation in the year 2009.

Following the methods described in Section 3.1 [22], for the current study we have extended the time series for NOx and $SO_2$ emissions per province up to the year 2018. The time series are based on annual averages to remove the seasonality and were normalised to the year 2005 for $SO_2$ and the year 2007 for $NO_x$. The results are presented in Figure 3, for $SO_2$ for the period 2005–2018 and for $NO_x$ for the period 2007–2018.

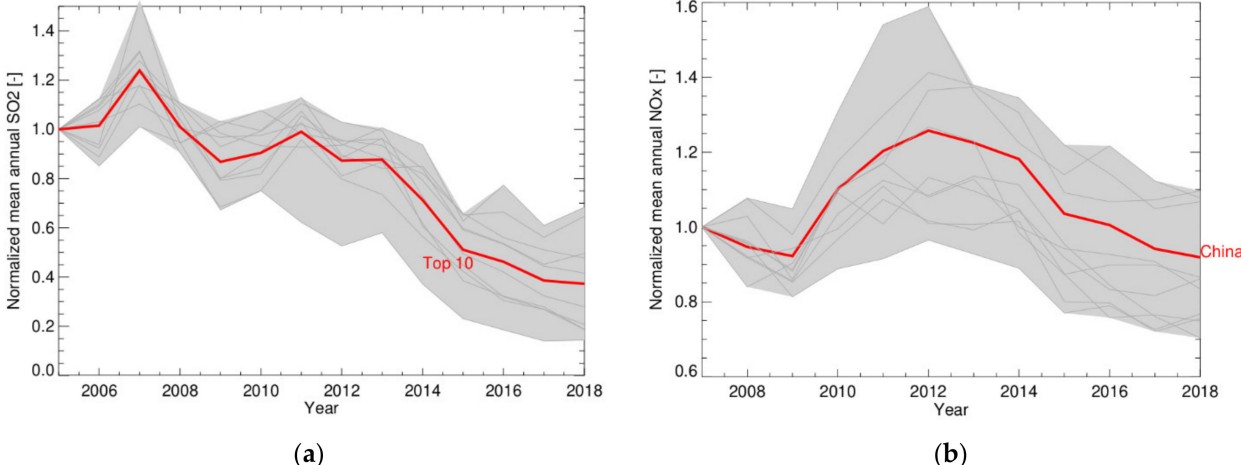

**Figure 3.** (**a**) Trends in SO$_2$ emissions (normalized to the reference year 2005). The grey lines (within the grey area) represent the individual trends of the 10 provinces with the highest SO$_2$ emissions. The red line shows the average of these 10 provinces. (**b**) Trends in NO$_2$ emissions (normalized to the reference year 2007). The red line is based on the average for East China.

Figure 3a shows the decline of SO$_2$ emissions after the year 2007, when desulphurisation was introduced and enforced for power plants and large industry. After 2013 the SO$_2$ emissions continued to drop considerably until 2018, when they seem to stabilize at a level of 40% of the emissions in 2005.

As China's power consumption and economy grew rapidly during the first decade of the 21st century, NO$_x$ emissions were also growing. However, as shown in Figure 3b, from 2012 onwards the NO$_x$ emissions gradually declined in response to air quality regulations related to traffic and NO$_x$ filtering at power plants [22,83]. Despite the strong growth of coal consumption, the economy and the number of cars in China, in 2018 the NOx emissions had decreased to a level similar to that in the period 2007–2009.

Identification of Emission Source Sectors

Emissions of sulfur dioxide (SO$_2$) and nitrogen oxides (NO$_x$ = NO + NO$_2$) over Eastern China were derived from OMI measurements of SO$_2$ and NO$_2$ columns to identify the underlying emission source sectors (industry, traffic, or nature) based solely on satellite observations. Because some source categories emit more SO$_2$ relative to NO$_x$ than others, the SO$_2$/NO$_x$ ratio in combination with absolute NO$_x$ levels dictates the emission source type. The results show the strong growth of the urban regions (with dominating anthropogenic emissions) in the period 2007–2017. A publication on this study is in preparation (Zara et al., in preparation).

BVOC Emissions from a Subtropical Forest

The BVOC emissions in the subtropical Pinus forest are dominated by monoterpenes (71.6%), with a contribution of 21.2% from isoprene. Diurnal and seasonal variations of the isoprene and monoterpene emissions were observed, with the daily maxima occurring after the highest PAR intensities were measured. The BVOC emissions were highest in the summer and lowest in the winter, as expected from the annual variation of air temperature and solar irradiation. Interannual variations of BVOC emissions were observed and the annually averaged BVOC emissions in 2015 were lower than in 2013. This is ascribed to the lower PAR, air temperature and water vapor concentration together with higher concentrations of trace gases and aerosols in 2015 (as indicated by the ratio of the diffuse to the global irradiation) [65]. The analysis of the data confirms that the driving factors for BVOC emissions are PAR and air temperature. The role of atmospheric constituents

(i.e., $O_3$, $NO_x$, $SO_2$, $H_2O$, aerosols, clouds) is in their effect on the direct and diffuse solar radiation reaching the canopy.

Using the BVOC measurements in the subtropical *Pinus* forest, an empirical model of BVOC emissions (EMBE) was developed based on considerations of the radiation balance near the canopy. The EMBE results were evaluated versus the data on different time scales and applied to simulate BVOC emissions from some other representative forests. Details on the BVOC emission measurements and the analysis of the data, including the EMBE, were published in Bai et al. (2017) [65].

Formaldehyde (HCHO) is a main product from BVOC oxidation [84] and its main losses are photolysis by UV radiation and reaction with OH radicals [76]. Relationships between the BVOC emission fluxes ($mg\ m^{-2}\ h^{-1}$) measured in the subtropical *Pinus* forest during the period January 2013 to December 2016 and monthly mean OMI-derived HCHO vertical column densities (VCDs, $molec\ cm^{-1}$) were determined [68]:

$$\text{isoprene emission} = 1.29 \times 10^{-16}\ C_{HCHO} - 0.77\ (r^2 = 0.86),$$

$$\text{monoterpene emission} = 1.02 \times 10^{-16}\ C_{HCHO} - 0.21\ (r^2 = 0.69),$$

$$\text{BVOC emissions} = 2.31 \times 10^{-16}\ C_{HCHO} - 0.98\ (r^2 = 0.80),$$

where $C_{HCHO}$ is the monthly mean HCHO VCD. Comparison of BVOC emission fluxes calculated using these relationships with EMBE model calculations show relative biases of less than 30%, in agreement with the uncertainties of BVOC emission flux measurements [68]. Similar correlations were also found for a temperate forest (42°24′N, 128°6′E), and a subtropical Lei bamboo plantation (30°18′N, 119°34′E) [68]. Therefore, the satellite HCHO VCDs provide a suitable method to obtain BVOC emissions on a large or regional scale.

Long-Term Variations and Mechanisms of Air Pollutants in North China

Ground-based measurements of concentrations of air pollutants (NO, $NO_2$, $SO_2$, $O_3$, $PM_{2.5}$) were made from May 2005 to January 2015 [55]. During these 10 years, the annually averaged surface concentrations of $NO_x$ and $SO_2$ decreased by 3.37% and 0.78% per year, respectively. In contrast, the annually averaged surface concentrations of $O_3$ and $PM_{2.5}$ increased during 2009-2014. These results were compared with time series of satellite-derived VCDs for HCHO, $NO_2$ and $SO_2$, at four representative sites in North China, from January 2005 to December 2015. The satellite data show that the annual mean values of the HCHO, $NO_2$ and $O_3$ VCDs and the AOD over the Luancheng, Xianghe, and Xinglong sites increased during 2005–2015, whereas the annual mean $SO_2$ VCDs decreased. These observations suggest a photochemical mechanism: the larger concentrations of $PM_{2.5}$ and $O_3$ are produced by the enhanced emissions of AVOCs and BVOCs and their reactions with OH radicals and other atmospheric constituents initiated by solar radiation at UV and visible wavelengths.

The analysis of the satellite data shows that annual mean VCDs of HCHO [85], $NO_2$ and $O_3$ at the 4 sites increased during most of the study period (2005–2015). The annual mean AOD increased over Luancheng, Xianghe, and Xinglong. In contrast, the annual mean $SO_2$ VCDs decreased.

The increase of the HCHO VCDs at the four sites indicates an increase of both AVOC and BVOC emissions. The largest increase of HCHO VCD was observed at Xinglong and is attributed to increased oxidation of BVOCs emitted from the local vegetation in the mountains. The forest coverage in Xinglong County in Hebei province is very high (66%). The increase of the HCHO VCDs at the other three sites is mainly caused by the increase of AVOC emissions. More details were presented in Bai et al. [68].

4.1.2. Conclusions

The OMI-derived $NO_2$ emissions, using the DECSO model, show the large variability across China, with the largest emissions in the area north of the Yangtze River and in the southwest, i.e., in the Sichuan basin. Hotspots are observed near the large cities and

industrial centers. Time series, extended here from the earlier work by van der A et al. [22] to include 2016, 2017 and 2018, show the decline of the concentrations from the start of the air quality regulations imposed by the Chinese government, for $SO_2$ in 2007, and for $NO_2$ in 2012. It is noted that the concentrations of both $SO_2$ and $NO_2$ in 2018 are close to those in 2017, suggesting that the decline comes to a halt. In 2017 and 2018 the $SO_2$ concentrations are reduced to about 40% of those in 2005; for $NO_2$ the concentrations in these years have decreased to a level similar to that in 2009. These observations show that the air quality regulations were indeed successful.

BVOC emission fluxes were measured in a subtropical *Pinus* forest during 2013–2016. The results were used to formulate an empirical model for BVOC emissions. The good relationships between monthly BVOC emissions and HCHO VCDs in 2013–2016 suggest further study of these relationships for the application of satellite data for emission estimates over large spatial scales.

Air pollution control measures were suggested, i.e., to reduce AVOC and human-induced BVOC emissions (due to cutting plants in cities, biomass burning, etc.) together with more strict NOx and $SO_2$ emission control [68].

*4.2. AEROSOL: Satellite-Derived Aerosol Properties over Mainland China: Application to Air Quality and Trend Analysis*

4.2.1. Results

Spatiotemporal Variation of Aerosols over China and Application in Air Quality Studies

A primary focus of the AQ-China aerosol project was the use of Aerosol_cci AOD data to present the current status of aerosol concentrations over China. To this end, ATSR data were used and, in view of the time coverage from 1995 until 2012, the data were extended both backward to 1987 using AVHRR data [40] and forward to 2017 [4,23]. The results were described in detail in the indicated references and will not be repeated here. Overall, the data show the strong variability of the AOD across China, the strong interannual and seasonal variations, and the decreasing trends during the period 2011–2017 after the initial rise until 2006.

In this paper we extend this study with another three years, 2018–2020, using the MA-IAC MODIS/Terra and MODIS/Aqua merged AOD product MCD19A2. As an example, a map of the annual mean AOD over China in 2018 is presented in Figure 4. The map shows the strong variability of the AOD, with very low values over the Tibetan Plateau (<0.1) and high values (up to 0.7 and locally even >0.9) in the eastern part of China where 94% of the population lives [86]. The strong anthropogenic pressure in the highly industrialized Eastern China, the high traffic density and domestic needs result in high AOD. The AOD is further augmented by weather conditions conducive to the formation of haze, the influence of long-range transport of dust from the deserts in the west and north of China (a natural aerosol source) and biomass burning aerosol (which can be of both anthropogenic and natural origin). In addition the geographical location and the occurrence of orographic effects plays a role as it affects transport of aerosol. All these factors together are responsible for the high AOD levels. Another area with high AOD (0.3–0.6, locally even >0.9) is the Taklamakan Desert in the west of China. The annual mean background AOD value is in the range of 0.1–0.2, which in 2018 was observed over a very large area along the northern border stretching from the east to the west.

The AOD over China shows strong spatial variability due to a variety of influences, as discussed in the Introduction section. Several studies were made to explain these variations in terms of transport pathways using meteorological information [12] and air mass trajectory analysis [44]. Wang et al. [44] used sun photometer data together with ground-based data from Nanjing during episodes with heavy pollution, in the summer and in the winter. These authors concluded that most of the pollution was of local origin with effects from regional transport (biomass burning aerosol) in the summer. During the winter episode, the aerosol concentrations were strongly influenced by long range transport of coarse particles (likely desert dust) from the arid areas in the west and north of China.

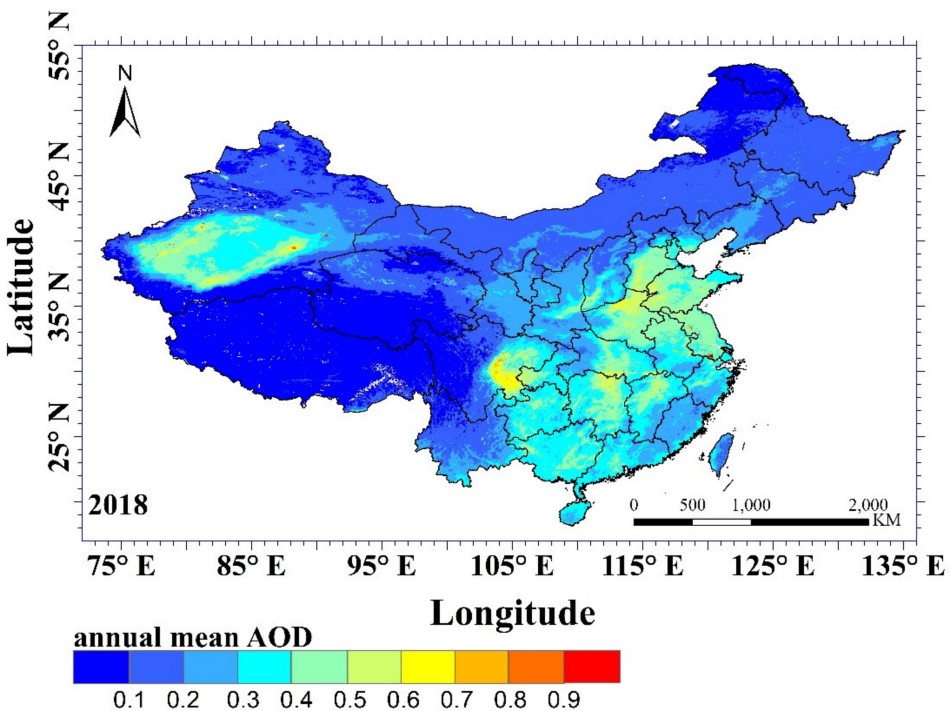

**Figure 4.** MAP of the annual mean AOD (at 550 nm) over China in 2018, produced using the MAIAC MODIS/Terra and MODIS/Aqua merged AOD product MCD19A2. The data were plotted with a spatial resolution of $1 \times 1$ km$^2$.

Hou et al. [12] characterized large scale weather patterns influencing the aerosol concentrations in the BTH and YRD in the winter, using the principal component analysis (PCA method). These authors divided the synoptic weather patterns in 4 types, according to the influence of the Siberian High and the northeast cold vortex, with different effects on the accumulation of air pollution and long-range transport.

To study the temporal variation of the AOD for the period 2011–2020, two highly industrialized regions were selected i.e., the regions around Shanghai (119.5–121.5°E; 30–32°N) in the Yangtze River Delta (YRD) and Zhengzhou (112–114°E; 34–35.5°N) in the Pearl River Delta (PRD). Monthly mean time series of the MAIAC MODIS/Terra and MODIS/Aqua merged AOD product MCD19A2 (at 550 nm) were used to produce annual mean series. Because of the strong seasonal variations of the AOD [4], the monthly mean time series were first de-seasonalized using a centered moving average with a period of 12 months, and then averaged to annual mean values to reduce effects of other influences than seasonal variations. The results in Figure 5 for Zhengzhou indeed show the decrease in the AOD between 2013 and 2016/2017 whereas it remained close to 0.5 thereafter. For Shanghai, an AOD peak is observed in 2014, with a decrease until 2018 and somewhat larger values in 2019 and 2020. Such fluctuations may reflect interannual variations due to, for instance, meteorological influences, which need to be further investigated. The main conclusion is that for AOD the decreasing trend seems to be halted in recent years. In both Shanghai and Zhengzhou, the AOD in the years 2018–2020 was on average about 65% smaller than in 2011 when the AOD was at a maximum. The value of the annual mean AOD averaged over all SE China in 2017 was close to that in 1995, as data in [39] show. In future work the temporal variations in these and other areas will be investigated in more detail.

It is noted that in both Shanghai and Zhengzhou, the AOD in 2020 was somewhat lower than in 2019. This may be due to reduced emissions during the COVID-19 lock-down period in early 2020, leading to lower aerosol concentrations over most of China (see, however, e.g., [26,87–89] showing the increased AOD over the NCP in response to changes in the oxidizing capacity due to the reduction of NO$_2$ emissions). Effects of the lockdown on the AOD and

the concentrations of several trace gases over China were reported in, e.g., Fan et al. [26] and Zhang et al. [45] and effects on $NO_2$ emissions were reported in Ding et al. [47].

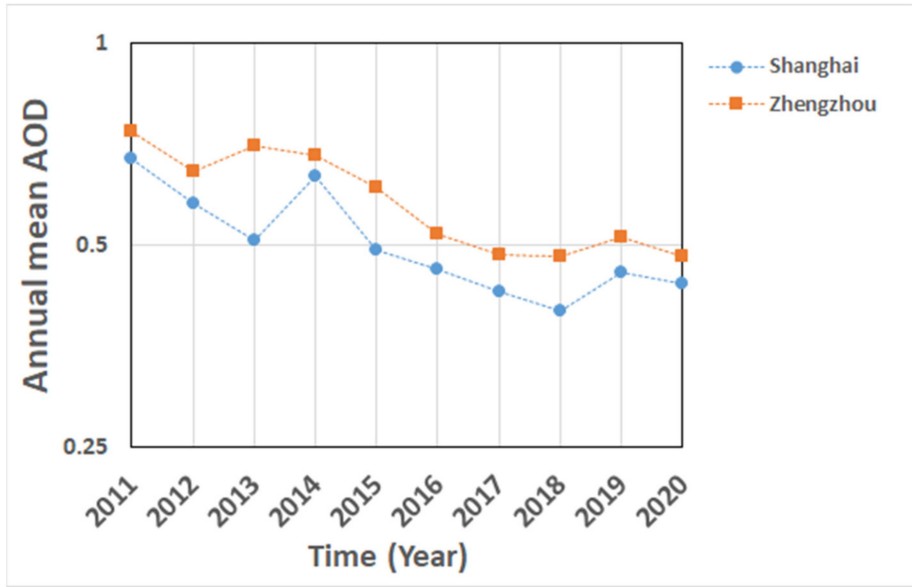

**Figure 5.** Time series of annual mean AOD (at 550 nm) over Shanghai and Zhengzhou, derived from the MAIAC MODIS/Terra and MODIS/Aqua merged AOD product MCD19A2, for 2011–2020. Note that AOD is plotted on a logarithmic scale. The annual mean values were calculated from de-seasonalized monthly values.

The use of AOD data for AQ applications requires the conversion of the column-integrated value of the AOD, an optical parameter, to the dry mass of the aerosol particles with an in situ aerodynamic diameter smaller than 2.5 μm ($PM_{2.5}$) near the surface. The $PM_{2.5}$/AOD relation depends on meteorological parameters. In the study of the spatiotemporal variation of $PM_{2.5}$ in the Guanzhong Basin [42], VIIRS AOD data were used as the main parameter together with meteorological parameters, land cover and population density, in a two-step LME-GWR model. Relations between $PM_{2.5}$ and the influencing parameters were derived and the model captured the $PM_{2.5}$ variation well, with better results in the summer and autumn than in the winter and spring. Annual averaged $PM_{2.5}$ concentrations in the Guanzhong Basin were higher than China's ambient air quality standard of 35 μg/m$^3$. $PM_{2.5}$ concentrations larger than 140 μg/m$^3$ were not well reproduced (for more detail, see Zhang et al. [42]).

The $PM_{2.5}$/AOD relation was addressed by Zhang et al. [2] using MODIS C5.1 AOD data in the $PM_{2.5}$ remote sensing (PMRS) model [90]. The PMRS model was improved by including the spatial distribution of the hygroscopic growth factor, based on measurements of the hygroscopic parameter (κ) at 17 locations across China, and Kohler theory. Another improvement was the use of the AOD fine mode fraction (FMF). The FMF is not provided in more recent collections of the MODIS aerosol data (after C5.1) and is not well correlated with AERONET FMF. However, the MODIS FMF was corrected to the AERONET FMF using a linear regression fit and the results compare favorably with PARASOL FMF data in the overlapping time period. With these improvements, the PMRS model calculated $PM_{2.5}$ values were in good agreement with ground-based observations: overall, 89% of the data pairs are within the error envelope, with the PMRS data somewhat smaller than the ground-based $PM_{2.5}$. With this good performance, the PMRS model was used to calculate the $PM_{2.5}$ over China for the years 2000–2015, showing the spatial and temporal variations. The data show a clearly decreasing trend in the years 2010–2015. Furthermore, contributions of anthropogenic and meteorological factors could be separated and used to

predict the evolution of $PM_{2.5}$ concentrations for different scenarios (for more detail, see Zhang et al. [2]).

In another study, Zhang et al. [77] used SONET sun photometer data to develop an improved inversion scheme for aerosol components. The new inversion scheme is based on the determination of the complex refractive index of a multicomponent liquid system for 7 aerosol components. The contributions of each component to the AOD were determined using a forward model for different combinations and minimizing the differences with the observations. This scheme was applied to all 16 SONET sites and the results show the spatial variation of the aerosol composition across China, as well as the seasonal variation. Time series since the start of SONET in 2012 show the interannual variation and effects of the China's environmental control policies. For more detail, see Zhang et al. [77].

As mentioned above, the AOD FMF is an important parameter in $PM_{2.5}$/AOD research. This is because, in pollution situations, fine mode aerosol particles usually dominate the optical properties measured at wavelengths used on satellite sensors and larger particles are not sampled by $PM_{2.5}$ monitors which are designed with an inlet with a 2.5 μm cut-off diameter. However, the FMF is not routinely available from current operational satellites. The convolutional neural network NNAero developed by Chen et al. [43] for the joint retrieval of AOD and FMF from MODIS data is a significant improvement to this situation. The validation of the NNAero FMF vs. independent AERONET data (not used for training the NN) shows that a significant fraction of the data (68%) are within the expected error, albeit that small FMF is overestimated and large FMF is underestimated. The analysis of this effect indicates that it is due to the wavelength-dependent aerosol absorption. NNAero can be used over both dark and bright surfaces, with good results and better coverage than the dark target and deep blue algorithms. As an example, the AOD and FMF retrieved with NNAero over Northern China, with a spatial resolution of $1 \times 1$ km$^2$, for one day (23 January 2019) was presented. The comparison with MODIS data on $10 \times 10$ km$^2$ shows that the spatial patterns of the DB and NNAero are similar but the 1 km resolution AOD and FMF maps from NNAero show spatial details which are not resolved in the coarser 10 km maps. The FMF provides additional information. For more detail, see Chen et al. [43].

The improved TS algorithm (ITS) for aerosol retrieval using data from the AHI sensor on the geostationary Himawari-8 satellite is based on the assumption that the ratio of the surface reflectance in different spectral bands does not change between any two scan times within an hour [91]. For the application with the ITS algorithm, new aerosol models were developed based on cluster analysis of AERONET sun photometer data. ITS was applied to retrieve the Aerosol Optical Depth (AOD) over eastern China and the results compare favorably with collocated reference AOD data at eleven sun photometer sites (R > 0.8, Root Mean Square Error (RMSE) < 0.2). Comparison with the H8–AHI official version 2.1 level 2.0 5 km AOD product [92] and the MODIS C6.1 10 km AOD data (DT and DB combined dataset, DTDB) shows the good performance of the ITS method for AOD retrieval with different observation angles (for more detail, see Li et al. [91]).

As an example of the application of the ITS algorithm to air quality studies, a sequence of 6 images, from 00:00 until 05:00 UTC (08:00 am to 01:00 pm local time) on 22 March 2021, is presented in Figure 6. This sequence shows the hourly evolution of the AOD as well as cloud cover over the study area. The data in each figure show the variability of the AOD across China, with high AOD over polluted areas, in particular over the BTH and its wider surroundings. In the morning, the AOD in the center of that region was very high and exceeded the threshold AOD in the ITS algorithm (set to 2.5; exceedance is indicated by the white area inside the high-AOD area). At 02:00 am UTC (10:00 am local time) the AOD had decreased to within the ITS threshold, but was still very high with values exceeding 1.4. The AOD continued to decrease to a value of about 0.8 in the afternoon (05:00 am UTC, i.e., 01:00 pm local time). Likewise, the AOD was very high over another region, identified as Hubei, and the AOD developed in a way similar to that over the BTH region. Other features are visible as well, such as the development of clouds over a large part of the study area in the early afternoon. The "lines" with elevated AOD are identified as occurring in

mountain regions and likely indicate valleys where aerosol accumulated. It is noted that the color scale in Figure 6 was chosen to include high AOD features, with a different choice also the evolution of lower AOD could be followed.

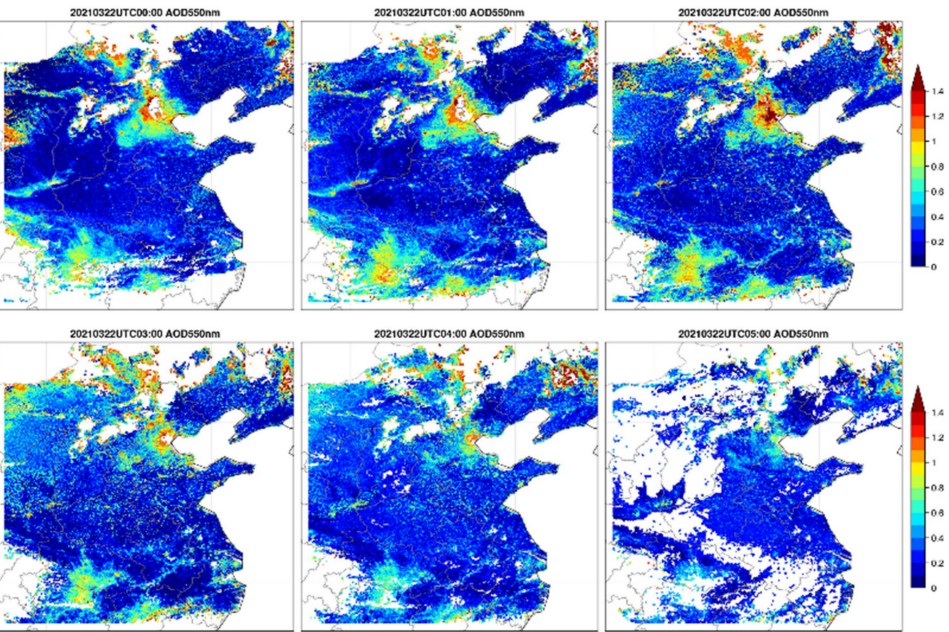

**Figure 6.** AOD retrieved over the study area using the ITS algorithm with data from the AHI sensor on the geostationary satellite Himawari-8, on 22 March 2021, from 00:00 until 05:00 UCT (08:00 am to 01:00 pm local time). The AOD values range from 0 to 1.8 as indicated in the color bars to the right. The data were plotted with a spatial resolution of 5 km. See text for further explanation.

Atmospheric Corrosion

Figure 7 shows the corrosion estimates for limestone (a), zinc (b) and carbon steel (c) materials and the soiling of modern glass samples (d) using DRFs with ground-based environmental data (red bars), the same estimates using DRFs with remotely sensed data (blue bars) and the Relative Approximation Error (RAE) between the two estimates for 10 European sites during different exposure periods.

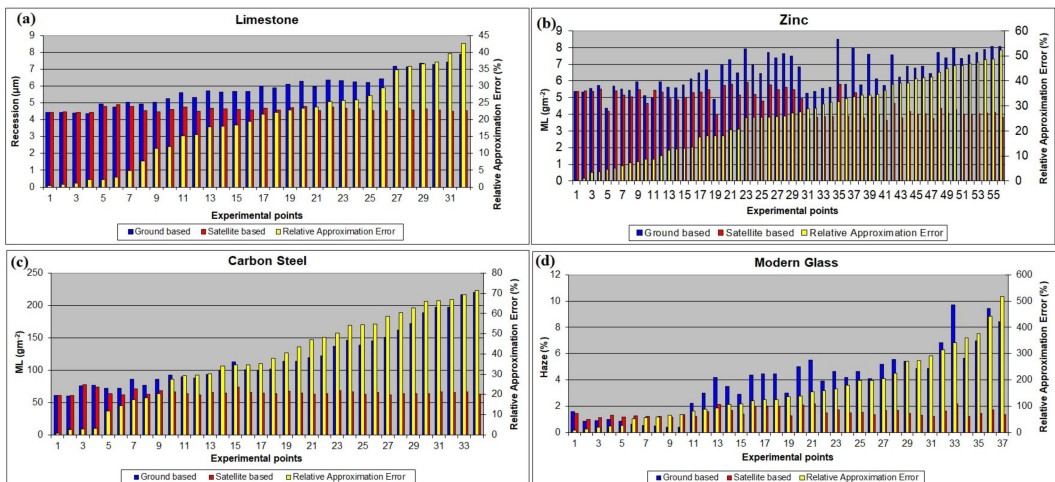

**Figure 7.** (**a**) Limestone recession estimates using appropriate DRF along with ground data (blue columns) and satellite data (red columns) for 10 European sites during different exposure periods. The relative approximation error (in %) between these two values is shown with yellow columns. (**b**) Same as (**a**) but for zinc. (**c**) The same as (**a**) for carbon steel. (**d**) The same as (**a**) for modern glass.

The data in Figure 7 show that in about 80% of the cases studied for limestone, the RAE is less than 30%. This result suggests the good applicability of the procedure proposed by Christodoulakis et al. [48] for this material. Further improvement in the $PM_{10}$ estimation procedure and the inclusion of $PM_{2.5}$ in DRFs is expected to increase the applicability. For zinc, in about 60% of the cases studied the RAE is less than 30%, indicating a relatively good application of the investigated procedure for this material. Further improvement of the procedure to estimate the effect of $SO_2$ is expected to enhance implementation. For carbon steel, in about 40% of the cases examined, the RAE is less than 30%. This indicates a limited applicability of the investigated procedure for this material. Further improvements in $SO_2$ and $PM_{10}$ estimation procedures and the inclusion of $PM_{2.5}$ in DRFs are expected to increase applicability. Finally, in the case of modern glass, in 70% of the cases considered, the estimated RAEs are greater than 85%. Further investigation of these large differences in the soiling results of this material is required.

### 4.2.2. Conclusions

The improvement of the aerosol retrieval algorithm for ATSR (ADV) as part of the ESA Aerosol_cci project resulted in an AOD data set of similar quality as that of MODIS, albeit with less coverage due to smaller swath width. Merging of the ATSR and MODIS AOD data sets resulted in a time series starting in 1995 (ATSR-2) to 2017. In this paper, a complementary MODIS dataset (MAIAC) was used to study the AOD over China in more recent years, i.e., the period 2010–2020. The spatial distribution in 2018 has features similar to those in previous years, but the annual mean time series for two selected regions, Shanghai and Zhengzhou, show that the AOD decrease seems to have come to a halt during the last couple of years. In both these regions, the AOD in the years 2018-2020 was on average about 65% smaller than in 2011. Averaged over all SE China, the annual mean AOD in 2017 was close to its value in 1995, as data in [39] show.

This time series was also extended backward to 1987, using AVHRR data. To this end, the RADI/CAS aerosol optical depth over land (ADL) algorithm for AVHRR [72] was further developed. The evaluation of the AVHRR-derived AOD versus AERONET data and versus AOD derived from solar radiation measurements, using the broadband extinction method (BEM), shows the good performance for low AOD, but improvements are needed for high AOD [40].

Maps of the spatial distribution of the AOD over China show the variability, with strong seasonal influences and meteorological effects on local, regional and synoptic scales. Meteorological effects are different in summer and winter [93]. Different synoptic situations in the winter result in different transport pathways and pollution episodes in the BTH and YRD [12].

The ITS algorithm for application to AHI/H8 geostationary data was successfully developed. The results show the diurnal variation of the aerosols over land [91]. A neural network (NNAero) was developed for the retrieval of AOD and FMF from MODIS data. The evaluation shows the good performance and better coverage than for the MODIS DT and DB operational products. In particular, the high resolution shows more detail than the MODIS data. FMF is a new product which currently is not operationally available from the common satellite data bases [43].

FMF is an important product for the application of AOD in AQ studies to provide $PM_{2.5}$. The correction of MODIS C5.1 data using a fit to AERONET FMF works well, as shown by comparison with PARASOL data for the overlapping period [94]. This correction was used in the PMRS model [90] together with an improvement to the hygroscopic growth factor. The $PM_{2.5}$ calculated using the updated PMRS model was evaluated versus ground-based data, with good results.

The use of AOD data to provide $PM_{2.5}$ was explored using VIIRS data in a two-step model. The comparison of the results over the Guanzhong Basin with ground-based $PM_{2.5}$ data show the good performance of the model [42].

An improved inversion scheme was developed for the retrieval of aerosol components from AOD data which was successfully applied to SONET data and show the variability of aerosol composition across China [77].

The main conclusion from the study of atmospheric effects on corrosion is that the proposed procedure [48] works well for limestone, with an RAE of less than 30% in about 80% of the cases. For other materials investigated, the number of good cases is less and further improvements are needed, especially with regard to the effects of gaseous pollutants and $PM_{2.5}$. The overall results of this study show that it is possible to use satellite data to monitor the corrosion of materials.

## 5. Overall Discussion

Air Quality is determined by concentrations of trace gases ($NO_2$, $SO_2$, $O_3$, CO) and aerosol mass concentration ($PM_{2.5}$ and $PM_{10}$) near the surface. In the AQ-China project the focus was on the use of satellite data for $SO_2$, $NO_2$ and AOD. Where available, tropospheric column densities were used, such as for $NO_2$. Yet, the interpretation of the satellite data to obtain information relevant for AQ studies is not straightforward. In this paper, OMI $NO_2$ and $SO_2$ data were used together with the DECSO model to determine the emissions of these trace gases over China. Maps showing the spatial distribution of emissions reveal hotspots associated with anthropogenic activities such as power plants, industrial centers near cities and along rivers, ship tracks, and other transport routes. $NO_2$ and $SO_2$ are trace gases with a relatively short atmospheric lifetime (several hours), as compared with aerosols with a residence time of several days to a week [95] (p. 443, Table 7.7). As a result, aerosol point sources are more difficult to detect because aerosol particles, with an atmospheric lifetime of several days to a week, are transported along with the wind and concentration patterns are more diffuse. Nevertheless, large industrial centers and cities may stick out during episodes with meteorological conditions conducive to the formation of haze, which may occur over extended areas such as frequently observed over the North China Plain. Also in the large basins such as the Sichuan basin and the Guanzhong basin, the air pollution frequently accumulates because there are no transport pathways facilitating dissipation of pollutants. Such situations in the NCP and YRD were investigated by Hou et al. [12] who defined 4 different types of synoptic situations during wintertime, in connection with aerosol pollution levels, accumulation and dissipation. In particular, the Siberian High and Cold Vortex are driving factors which determine the occurrence of pollution or clean air in the NCP and YRD. For the Guanzhong Basin, Zhang et al. [45] used a two-step transport model to determine $PM_{2.5}$ concentrations from VIIRS AOD data and the comparison with ground-based observations shows the good performance of the model.

$NO_2$ and $SO_2$ are precursor gases for the formation of aerosols and therefore spatial patterns of aerosols would be expected similar to those of $NO_2$ and $SO_2$. However, gas-to-particle conversion produces very small particles (nm) and it takes time to grow to sizes which can be detected by instruments used on satellites [54] or contribute to $PM_{2.5}$. In this time, the particles may have been transported with the wind and thus the aerosol concentration (or AOD, used as a proxy) distribution is more diffuse than the concentrations of the localized pre-cursor gases. In addition, directly emitted aerosol particles, such as biomass burning, dust or sea spray aerosol, contribute to the aerosol load and thus the aerosol spatial distribution. This was clearly shown in the study by Wang et al. [44] with different aerosol patterns in Nanjing in the summer and winter, with different local contributions, regional contributions from (agricultural) biomass burning in the summer and long-range transport of desert dust in the winter.

Other gases contributing to secondary aerosol formation are volatile organic compounds (VOCs) from anthropogenic (AVOCs) or biogenic (BVOCs) origin. VOCs contributing to new particle formation are large molecules which cannot be retrieved from satellite measurements. However, formaldehyde (HCHO) is formed in the photochemical oxidation of VOCs [96] and can be detected by satellites, such as OMI and TROPOMI [85].

The BVOC flux measurements in the sub-tropical *Pinus* forest were analyzed in terms of OMI-retrieved HCHO and appear to correlate well. In this analysis monthly averaged HCHO VCDs and BVOC emission fluxes were used which improves the correlation and reduces observational noise. Relationships were derived which are proposed to be tested for their application to other areas and on a larger scale.

Long-term satellite-derived time series are available for $NO_x$, $SO_2$ and AOD, and in the AQ-China project they were extended with three more years, i.e., for $NO_x$ and $SO_2$ emissions from 2015 in Van der A et al. [22] to 2018 and for AOD from 2017 [39] to 2020. Each of these time series shows the effect of the air quality regulations imposed by the Chinese government: for $SO_2$ the emissions started to decline in 2007, for $NO_x$ in 2012 and maximum AOD was observed in 2011. However, in the last year included in the current study, 2018, the $NO_x$ and $SO_2$ emissions did not decrease as fast as in previous years, suggesting that the decline may be slowing.

In a recent study [97] annual mean $NO_2$ time series over 11 regions distributed across China from north to south and from east to southwest, were presented for 2011–2019. These time series indicate that the decreasing trends in tropospheric $NO_2$ vertical column densities have halted in recent years, with differences in regions north and south of the Yangtze River. South of the Yangtze River, the decrease stopped around 2016, and concentrations in later years were within 10% of those in 2016. In the North China Plain (NCP) the decrease appears to have only halted in 2019.

Because $NO_2$ and $SO_2$ are precursor gases for the formation of aerosol particles, the reduced emissions could affect the aerosol formation and thus the concentrations. AOD time series constructed as annual averages over Shanghai and Zhengzhou, for the years 2011–2020 (Figure 5) indeed show the decrease of the AOD during the last decade, as reported also in Sogacheva et al. [39]. However, the extended period in the current study shows that over Zhengzhou the AOD was close to 0.5 in each of the years 2018–2020. Likewise, the AOD over the Shanghai region decreased until 2018 whereas in 2019 and 2020 it was somewhat higher. These observations lead to the conclusion is that for AOD the decrease has halted in recent years. In both Shanghai and Zhengzhou, the AOD in the years 2018–2020 was on average reduced to about 65% of the value in 2011 when the AOD was at a maximum [39]. In 2017, the annual mean AOD averaged over all SE China in 2017 was close to that in 1995, as shown from the data presented in [39].

In future work the temporal variations in these and other areas will be investigated in more detail. It is noted that the reduction of emissions of $SO_2$ and $NO_x$ may not be the only reason for the decline in the AOD. The time series of AOD show fluctuations and such interannual variations may be due to, for instance, meteorological influences. Both natural and anthropogenic effects influence the evolution of the emissions and concentrations. The influence of each of these on the satellite-observed concentrations of $SO_2$, $NO_2$ and CO and AOD were estimated for a period of about 10 years ending in 2013 [1]. Using the improved PMRS model with MODIS data to estimate effects of anthropogenic and meteorological factors on $PM_{2.5}$ concentrations in two scenarios [2] leads to the conclusion that the decrease of $PM_{2.5}$ concentrations after 2011 is mainly due to pollution control measures whereas effects of meteorological factors on $PM_{2.5}$ have increased. This study included data until 2015 and the modelled $PM_{2.5}$ concentrations for 2016-2018 compare favorably with measurements in these years. However, Zhang et al. [2] also used the model to extrapolate to 2025 and concluded that the anthropogenic emissions need to be further reduced to reach the target for $PM_{2.5}$ concentrations in 2025 of 35 $\mu$g m$^{-3}$.

An important parameter in the PMRS and other models used to estimate the concentration of $PM_{2.5}$ from satellite data is the AOD FMF. FMF is currently not available from operational satellite observations. Chen et al. [43] explored the use of a convolutional neural network for the retrieval of both AOD and FMF from MODIS data (NNAero). The network was trained with MODIS and AERONET data and the results were validated versus an independent AERONET data set (not used in the NN training). The validation

and the comparison with MODIS data show the better performance of NNAero, as regards coverage, spatial resolution and, in particular, the retrieval of FMF.

The ITS algorithm developed for AOD retrieval over land using data from the AHI sensor on the geostationary satellite Himawari-8 [91] is available to process images over large areas with high temporal resolution. Hourly time sequences show the evolution of AOD and cloud cover over mainland China.

Satellite data were used in studies on the corrosion of four different materials due to effects of air pollution and other environmental parameters. These effects were measured using the Dose Response Functions (DRF) which express the relationship between the corrosion or deterioration of a material after one year of exposure and the levels or loads of pollutants in combination with climatic parameters. The materials used are limestone, zinc, carbon steel and modern glass, and the DRFs for these materials were evaluated using either ground-based or satellite data. The comparison of the results shows that the satellite method works well for limestone, with an RAE of less than 30% in about 80% of the cases. For other materials investigated the number of good cases is smaller and further improvements are needed, especially as regards effects of gaseous pollutants and $PM_{2.5}$.

## 6. Main Conclusions

- Satellite-derived data sets for $SO_2$ and $NO_2$ emissions were extended up to 2018; spatial distributions show the hotspots in industrialized and urban centers near large cities and rivers.
- Time series of annual mean emissions of $SO_2$ and $NO_x$ per province show their decline during the last decade. However, the emissions of both $SO_2$ and $NO_x$ in 2018 are close to those in 2017, suggesting that the decline comes to a halt. This is confirmed by studies over smaller regions across China [97].
- In 2017 and 2018 the $SO_2$ emissions were reduced to about 40% of those in 2005; for $NO_x$ the emissions in these years have been reduced to a level similar to that in 2009.
- Existing AOD data sets were extended with three years and instead of the AATSR and MODIS C6.1 AOD data, the MAIAC MODIS/Terra and MODIS/Aqua merged AOD product MCD19A2 was used to reproduce a time series covering the period 2011–2020. The spatial distribution shows features similar to those in previous years with large spatiotemporal variations.
- The time series of the annual mean AOD over Zhengzhou and Shanghai show the decline of the AOD during the last decade. However, in each of the years 2018–2020 the AOD over Zhengzhou has a similar value, close to 0.5. Likewise, the AOD over the Shanghai region decreased to a minimum of 0.4 in 2018, whereas in 2019 and 2020 it was somewhat higher. These observations lead to the conclusion that for AOD the decrease seems to have halted in recent years.
- In both Shanghai and Zhengzhou, the AOD in the years 2018–2020 was on average reduced to about 65% of the value in 2011 when the AOD was at a maximum. In 2017, the annual mean AOD averaged over all SE China in 2017 was close to that in 1995, as shown by data in [39].
- AVHRR AOD over small areas in China and Europe were used to extend the AOD time series backward from 1995 (ATSR-2) to 1987 using the RADI/CAS aerosol optical depth over land (ADL) algorithm. This algorithm provides good results, except for the highest AOD where improvements are needed.
- The convolutional neural network NNAero for the joint retrieval of AOD and FMF from MODIS data is an improvement over the operational MODIS data sets, in particular as regards the retrieval of FMF.
- The ITS algorithm developed for the retrieval of AOD from AHI on the Geostationary satellite Himawari-8 can be used to study the diurnal variation of aerosols.
- An improved inversion scheme for the retrieval of aerosol components was developed and applied to all SONET sites across China, showing the spatial variation of the aerosol composition, as well as the seasonal variation.

- For the application of satellite observations in AQ studies, the AOD/PM$_{2.5}$ relation has to be evaluated. The application of a two-step model over the Guanzhong Basin shows good results as concluded from the evaluation versus ground-based PM$_{2.5}$ data.
- The PMRS model was improved by accounting for the variation of the hygroscopic growth factor across China and the inclusion of the FMF fraction based on a correction to MODIS data.
- The aerosol concentrations are strongly influenced by both anthropogenic and natural effects. The study by Kang et al. [1] shows the contributions of each of these factors to the concentrations of NO$_2$, SO$_2$, CO and to AOD.
- The different meteorological conditions during air pollution episodes in the summer and winter in Nanjing strongly influence the contributions of regional (summer) versus long-range transported (winter), adding to the locally produced pollutants.
- Synoptic situations in the wintertime determine the transport pathways resulting in the occurrence of either accumulation or dispersion of aerosol in the BTH and the YRD.
- A strong correlation has been observed between monthly mean BVOC emissions measured in a sub-tropical *Pinus* plantation and HCHO VCDs. The relationships are suggested to be used for the application of satellite data for emission estimates over other areas and over large spatial scales.
- The emissions of AVOC and human-induced BVOC (due to cutting plants in cities, biomass burning, etc.) need to be reduced, together with more strict NO$_x$ and SO$_2$ emission control [68].
- The study of the atmospheric effects on the corrosion of materials shows that the method developed for the use of satellite data works well for Limestone. Other materials require further improvements, especially in terms of the effects of air pollutants.
- The application of the PMRS model and factorization of natural and anthropogenic factors to forecast their effects until 2025 shows that stricter regulation of the anthropogenic emissions is needed to reach the targeted PM$_{2.5}$ concentration of 35 $\mu$g m$^{-3}$ in 2025.

**Author Contributions:** Conceptualization, G.d.L. together with all co-authors; methodology, all co-authors; software, R.v.d.A., J.B., C.F., J.D., I.C., G.K., X.C., X.H., J.W., K.Z., Y.Z., M.Z.; validation, all co-authors; formal analysis, all co-authors; investigation, all co-authors; resources, G.d.L., R.v.d.A., J.B., Y.X., C.V., Z.L.; data curation, J.B., C.F., J.D., I.C., G.K., X.C., X.H., J.W., K.Z., Y.Z., M.Z.; writing—original draft preparation, G.d.L., R.v.d.A., J.B., C.V., C.F.; writing—review and editing, all co-authors; visualization, all co-authors; supervision, G.d.L., R.v.d.A., J.B., Y.X., C.V., Z.L.; project administration, N/A; funding acquisition, G.d.L., R.v.d.A., J.B., Y.X., C.V., Z.L. All authors have read and agreed to the published version of the manuscript.

**Funding:** The work presented in this paper contributes to the ESA/MOST cooperation project Dragon 4, Topic "Atmosphere, climate & carbon cycle", sub-topic "Air-Quality". The Marco Polo project supported by the EU FP7 SPACE Grant agreement no 606953 contributed to the Dragon 4 project results as well as the Aerosol-cci project supported by ESA-ESRIN project AO/1-6207/09/I-LG. The work by CUMT was supported by the National Natural Science Foundation of China (NSFC) under grant no. 41871260. Contributions by LAGEO were supported by the Strategic Priority Research Program of the Chinese Academy of Sciences (grant no. XDA19070202) and the National Natural Science Foundation of China (grant no. 41275137).

**Institutional Review Board Statement:** Not applicable.

**Informed Consent Statement:** Not applicable.

**Data Availability Statement:** Data used in the reported studies were obtained from websites as indicated in the text.

**Acknowledgments:** The authors acknowledge the data providers for their ceaseless effort to collect, process, quality control and provide data of the best possible quality. This includes satellite data by the various space agencies, data providers and algorithm developers, as well as ground-based data

from a variety of networks and individual institutes. The authors thank the editor and anonymous reviewers for their time to read the manuscript and provide helpful suggestions.

**Conflicts of Interest:** The authors declare no conflict of interest. The funders had no role in the design of the study; in the collection, analyses, or interpretation of data; in the writing of the manuscript, or in the decision to publish the results.

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
