# Peer review of "Air Quality over China"

_remotesensing, doi:10.3390/rs13173542_

Round 1

Reviewer 1 Report

please see attached review.

Author Response

see attached file: remotesensing-1219589-Response to Reviewer#1_final

Reviewer 2 Report

Review of the paper: " Air Quality over China ", by G. de Leeuw et al. (Paper submitted to Remote Sensing, #remotesensing-1219589 )

Major comments:

1- Despite the interest that this kind of studies could present regarding the monitoring and management of the Air Quality in China, the paper suffers nevertheless from the necessary rigorousness regarding the methodology, the analysis and the presentation of the results.

2- The abstract does not reflect all what was presented in the paper: i) No quantification regarding the determination of trends in the concentrations of aerosols and trace gases, ii) No mention regarding the estimates of NOx, VOC, SO2 emissions as well as their variability; iii) the discrimination of local effects and those due to long range transport is not mentioned, iv) The effect of haze on air quality is mentioned in the abstract but it is not discussed in depth in the paper.

3- The introduction is too long and lacks many references in several places, so the scientific problematic in relation to the state of the art is not very clear. The motivations of the study are very confusing especially with respect to the use of all the datasets presented in the paper. We also note that there are many acronyms, most of which are not defined. I suggest that the authors define each acronym when it is first presented.

4- The methodology concerning the separation of the different effects that influence the air quality over China (meteorological factors, natural and anthropogenic emissions) is not developed as it should be. The authors could take advantage of the modeling part, which is not well developped, to illustrate well the quantification of each effect.

5- The paper uses a considerable dataset, however most of it lacks information about the used version, the spatial resolution and whether it is L2 or L3 (CALIOP, MODIS, VIIRS, …), there is also a lack of articulation between the different in-situ data used, e.g. AERONET, SONET, CARSNET.

6- The modeling part lacks a lot of details about the version of the model used, its input parameters, its spatial resolution, ...etc as well as how the results are evaluated and interpreted; I would have liked this part to be much more explicit in order to give added value to the paper in synergy with the different datasets used.

Minor comments :

1- P78: "evaluate the effects of aerosols", Which ones?

2- P153: P. R.?

3- P156: EO?

4- P178: aerosol optical depth: to be defined before (same remark on all acronyms: see comment 3 of major comments)

5- In table 1 : please define the version of the used quantities and indicates whether it is L2 or L3.

6- P295: more information about CHIMERE are welcome (which domain, resolutions, meteorological focings, period)

7- P321: EMBE, PAR, HOBO?

8- P372: any comment about this comparison?

9- P392: EDR? QF? SONET?

10- P413: 'The CNN was trained ...': over which period? Which domain? Temporal and spatial resolutions?

11- P419: CARSNET?

12- P436:'Atmospheric Corrosion Athens Station': what is the articulation of Athens station with the air quality over China?

13- P443: Any more information about the corrosion modeling techniques will be welcome

14- P492: please give more explanations about the SO2/NOx ratio

15- Figure4: resolution of the MODIS data? L2 or L3? Version?

16- Figure5: It is not clear where the AOD comes from and at what wavelength. It would also be nice to add the error bars to get an idea about the inter-annual variability

17- P648: Please give more details about the PM2.5/AOD ratio

18- P652: LME-GWR?

19- Figure 6 is not clear: title, colorbar. Could you indicate the spatial resolution of the measurements?

20- Figure 7 is very difficult to interpret

21- P763: it is not clear how this merge between ATSR and MODIS is done in terms of AOD: how the bias and possible differences between the two datasets are managed?

22- P774: Same remark for AVHRR

23- P884 : any further comment about the meteorological influences  ?

Author Response

remotesensing-1219589-Response to Reviewer#2_final

Reviewer 3 Report

The paper “Air Quality over China” is an overview of the comprehensive research activities done during the implementation of Dragon 4 project, following the involvement of a lot of resources (ground based and satellite instruments, modelling, and synergies between several techniques) to fulfil the objectives of the project.

The paper sounds relevant for the region and it covers all areas to characterise the trends in the concentrations of aerosols and trace gases, quantify of emissions using a top-down approach and gain a better understanding of the sources, transport and underlying processes contributing to air pollution.

I recommend the publication in its current form.

Minor comment: page 2 line 94- “Whereas” instead of “where”

Page 23 line 984- “PM2.5” instead of “M2.5”

Author Response

remotesensing-1219589-Response to Reviewer#3_final

Round 2

Reviewer 1 Report

Please see attached review. 

Author Response

This 2nd review was made while our response to the first review was not available to the reviewer. Hence it was not based on the information we provided and therefore a response to review 2 is not useful. 

Round 3

Author Response

see attached response
